# Healthcare Leadership Perspectives on Supporting Frontline Workers in Health Center Settings during the Pandemic

**DOI:** 10.3390/ijerph19063310

**Published:** 2022-03-11

**Authors:** Carmen Alvarez, Holly Sims, Kimesha Grant, Jessica Walczak, Paula Darby Lipman, Jill A. Marsteller, Lisa A. Cooper

**Affiliations:** 1Johns Hopkins School of Nursing, Baltimore, MD 21205, USA; hsims3@jhmi.edu (H.S.); klinton8@jhmi.edu (K.G.); jwalcza2@jhu.edu (J.W.); lisa.cooper@jhmi.edu (L.A.C.); 2Johns Hopkins Center for Health Equity, School of Medicine, Johns Hopkins University, Baltimore, MD 21205, USA; jmarste2@jhu.edu; 3Westat, Rockville, MD 20850, USA; paulalipman@westat.com; 4Department of Health Policy and Management, Johns Hopkins Bloomberg School of Public Health, Baltimore, MD 21205, USA; 5Division of General Internal Medicine, School of Medicine, Johns Hopkins University, Baltimore, MD 21205, USA

**Keywords:** COVID-19, healthcare leadership, healthcare workers

## Abstract

Throughout the COVID-19 pandemic much attention has been given to addressing the needs of hospital-based healthcare professionals delivering critical inpatient care. At the same time, other groups of essential frontline healthcare workers have continued to serve low-income and underserved populations whose healthcare and nonmedical needs did not cease, and in many cases were exacerbated by factors associated with the pandemic shutdown. As these same factors also potentially impacted well-being and effectiveness of frontline healthcare workers, we sought to understand the organizational-level responses to the pandemic, including the support and preparation for frontline workers. As part of a larger study focused on reducing health disparities in hypertension, we conducted semi-structured individual interviews with 14 leaders from healthcare and health services organizations to explore how these organizations responded to accommodate frontline workers’ needs. Findings from our sample show that healthcare and health service organizations made a range of major and timely modifications to clinic operations intended to address the needs of both employees and patients and strove to ensure continued patient services as much as possible. Nevertheless, our findings underscore the need for more attention and resources to support healthcare workers in primary care settings especially during emergencies such as COVID-19.

## 1. Introduction

Community health centers (CHCs) are the largest providers of primary care services nationwide, serving over 29 million people [1], mostly very low-income populations with greater healthcare needs than the general low-income population [2]. The effects of the pandemic were particularly challenging for communities served by these health centers, amplifying the health disparity outcomes they already aim to address. People of color, non-English speakers, undocumented patients, and persons living at or below the poverty level—including members of those same communities working as frontline workers—were devastated by the impacts of COVID-19 [3]. These populations had higher rates of COVID-19 infection and morbidity, and face disproportionate challenges at home, such as the lack of childcare and loss of family income [3,4,5].

Like many other health service providers, this critical safety net has been impacted by the COVID-19 pandemic [6]. Amidst their multiple challenges, committed essential workers—the CHC clinicians and staff—delivered services, often at personal risk to themselves. Although the need for services remained, COVID restrictions impeded access to care and challenged CHCs to modify services both to meet patients’ needs and to remain in service. Despite the multiple challenges the CHCs faced, including the lack of a national response plan, limited resources such as personal protective equipment (PPE), workforce [7], and capacity for telehealth [8], the CHCs still managed to play a critical role in COVID testing and delivery of quality care [1].

There is a paucity of research that integrates the needs of healthcare workers serving vulnerable populations into emergency operations and preparedness planning [9]. Despite their importance and essential role to the nations’ medically underserved, little is known about how individuals providing primary care services to high-need and marginalized communities were supported during the pandemic. We do know that clinician burnout was a crisis prior to COVID-19 [10] and has been amplified with the pandemic [11]. At the start of the pandemic (March 2020), The Larry A. Green Center conducted a progressive series of national surveys to evaluate the status of primary care centers and healthcare worker perspectives [12]. Survey findings from primary care clinicians revealed multiple concerns including staffing shortages and challenges with handling patient volumes. In addition, other issues such as fear of contracting and transmitting the virus to loved ones and increased workloads due to personnel shortages exacerbated burnout among healthcare workers [13].

The CHC leadership has a major role in caring for healthcare workers for both their well-being and for supporting the delivery of quality care. In response to the pandemic, attention has been directed toward increasing workforce capacity through activities such as maximizing the scope of practice (i.e., removing certain requirements for select clinicians and other health professionals in order to facilitate efficiency in care), increased flexibility on licensing regulations, and interstate reciprocity [14].

In sum, despite the high priority of maintaining the provision of essential health services during the pandemic, how this workforce was supported to optimize functioning and manage stressful workloads has received less attention. Helping to prevent worsening of health disparities from COVID-19 and other social determinants of health will require support from interdisciplinary healthcare teams that function at the highest level and that address both medical and nonmedical needs. The purpose of this study was to understand and describe how organizational healthcare leaders have supported frontline workers in ambulatory settings during the pandemic.

## 2. Materials and Methods

### 2.1. Study Design and Procedure

The Johns Hopkins School of Medicine University Institutional Review Board approved all study procedures (IRB00085630). This qualitative study was part of a larger pragmatic trial, The RICH LIFE Project, that started pre-pandemic [15]. The trial was designed to compare the effectiveness of enhanced standards of care with a clinic-based, multilevel intervention that included behavioral change counseling by nurse care manager and, if needed, additional support from a community health worker to improve blood pressure control. The trial included partnerships with several federally qualified health centers (FQHC) in urban and rural areas across Maryland and Pennsylvania, and a healthcare organization with private clinics in urban and suburban locations. Although a private organization, this organization’s clientele included populations with low-income. The study also included a partnership with an organization that did not provide clinical services, rather their expertise was providing outreach services (i.e., food and housing assistance) primarily to underserved populations. Toward the end of the trial (fall 2020), exit interviews were conducted with organizational leaders to understand their experiences with the implementation and engagement of the project protocols. Given the timing, questions about how their organizations responded to the pandemic and how they were supporting their staff were included in the interview guide.

Twenty-three organizational leaders representing five different healthcare organizations (four federally qualified health centers and one private organization that serve a low-income population) and the two health services organizations were invited via email to participate in the exit interviews. Fourteen leaders consented to participate in the extended interview that addressed responses related to COVID workplace adaptation. These interviews were conducted by researchers from the coordinating center for the study program. All interviews were conducted virtually via WebEx. The interviewers used a semistructured interview guide to facilitate discussions about the leaders’ experiences during COVID regarding clinic operations and supporting their staff members and clients. All participants (healthcare leaders) consented to being audio-recorded; these recordings were then transcribed verbatim, deidentified, and reviewed for accuracy. Interviews were on average 36 min (28–51 min) in duration and produced on average 13 pages of text (9–21 pages).

### 2.2. Data Analysis

NVivo12 was used for data management and to facilitate the inductive data analysis. Four team members individually read samples of the interviews and developed preliminary codes and themes that emerged from the data. The team then met to compile and discuss the codes to refine the codebook for focused coding of all interviews. Each interview was then dually recoded using the finalized codebook. NVivo was used to assess inter-rater reliability, and discrepancies were resolved as a group. Throughout the coding process, the team met to discuss emerging themes.

## 3. Results

Collectively organizational leaders described a stressful time requiring flexibility in response to situations in which information changed daily and sometimes multiple times a day. They reported focusing on the need to make decisions regarding restructuring clinic operations to be compliant with COVID restrictions but also ensuring that patients’ needs continued to be met. The leaders also acknowledged the increased stress experienced by their healthcare workers and the limited options available for alleviating their stress. These findings are organized under three themes—restructuring clinic operations, stressed frontline workers, and organization support for frontline workers—as discussed below. Table 1 highlights these themes and provides illustrative quotes that emerged from the data.

### 3.1. Restructuring Clinic Operations

Healthcare leaders were placed in a challenging position in which they needed to maintain basic clinic functioning, while at the same time modifying operations to keep up with the rapidly evolving pandemic. The study participants recalled the early days of the pandemic when immediate action was needed to protect workers and patients. Several leaders established executive teams within their organization to plan and implement changes to clinic operations. The leadership endeavored to follow the latest guidelines on safety protocols as indicated by state or federal mandates. The ever-changing guidance, however, frequently caused frustration as the healthcare leaders attempted to keep up with the latest protocols. In the words of one leader: “*The number of changes in the beginning was huge! And nonstop. Even multiple changes on any given day. You know, ‘Wear masks,’ ‘Don’t wear masks.*” [FQHC 1, Director] Another leader shared: “*We would get information at eight o’clock in the morning, and by two o’clock something had changed, and so we rapidly had to make a sudden and quick decision on what to do with that change*.” [FQHC 5, Director].

Despite the changing guidance during the early days of the pandemic, most organizations rapidly established safety protocols and modifications to clinic operations in order to limit COVID-19 exposure as much as possible. New operations included allowing some staff members to work from home, ceasing clinic services that were deemed nonessential, closing practice sites, and moving certain services outdoors or to modified environments. Leaders voiced that their decisions were often made based on the needs of both the patients and their workers, while also considering the need to maintain financial solvency and basic operations. Several organizations were able to allow all staff members to work from home, while other organizations enabled remote work only for certain workers depending on their specific role or level of risk. Organizations that could not accommodate remote work for all staff members established protocols that allowed for alternate scheduling and decreased face-to-face contact.

The modifications to clinic operations presented challenges to the leadership, particularly regarding financial solvency and the ability to provide services. While the leaders acknowledged that new operations were necessary to ensure worker and patient safety during a time of crisis, several voiced their concerns regarding the financial impacts. The shut-down impacted patient volume and, consequently, revenue. This forced leaders to make decisions about how to adjust staff roles and responsibilities to continue essential clinic operations. In preparation for resuming on-site activities, some leaders mentioned conducting training for proper use of personal protective equipment.

*Implementation of telemedicine*. A universal modification was the establishment of telemedicine services in order to decrease the number of patient clinic visits or to stop in-person appointments altogether. Some clinics switched entirely to telemedicine, ceasing all in-person visits; others adopted a hybrid approach with both on-site and video (or telephonic) visits. Some clinics offered both modalities to the patients; patients were triaged as indicated by clinic policy. In other clinics, patients seeking preventive or low-acuity services were more typically seen remotely, and patients with more urgent needs were seen in-person. Other clinics chose to serve only healthy patients for in-person visits to prevent high-risk or elderly patients and employees from being exposed to the virus.

The use of telemedicine required clinic staff to learn new systems and undertake additional responsibilities. Several healthcare leaders reported that telephone calls increased significantly during the pandemic, which necessitated restructuring of clinic staff to accommodate the increased call volume. This included the implementation of a phone triage system whereby nursing staff were assigned to determine whether particular patients should be seen via telemedicine or in-person based on their symptoms or needs. This use of telemedicine created other challenges for patients, particularly those with chronic illnesses, e.g., the lack of medical equipment at their homes made it difficult to monitor and manage their conditions. Many patients experienced challenges with connectivity. For example, one leader shared: “*You know, many of our patients are tech savvy*. [But] *Many of them are not.… Sometimes it had taken us, in the beginning, an hour or more to get a patient on a video visit*.” [FQHC 1, Director] Notwithstanding these challenges, the healthcare leaders identified some benefits of telemedicine including that it allowed for a reduction in in-person visits, more remote work, and as a result decreased the need for expenses on other supplies such as personal protective equipment or cleaning supplies.

*Keeping up with patients’ needs*. Many of the decisions made in the rapidly changing context of the pandemic were centered on the evolving needs of patients. Respondents enumerated the challenges facing their patient population pre-pandemic and how conditions were exacerbated, such as the lack of food and housing access and medication assistance. These challenges were also compounded for those who lost employment due to the pandemic. Healthcare leadership made the effort to modify services to address some of their patients’ needs, including calling and checking more frequently on seniors who lived alone and delivering food and medications to patients. For example, one leader explained:
*We actually had our care managers deliver food to their [patients’] home, put it on their front porch. Go to the pharmacy and pick up medications for them, which is really not something that we would do in our normal day-to-day routine but it’s something that needed to be done.*[Healthcare organization 2, Manager]


More directly, in response to the pandemic, some health organizations distributed masks and hand sanitizer in the community and collaborated with community partners to provide COVID testing at schools and places of faith.

### 3.2. Stressed Frontline Workers

Indeed, clinicians and staff were the ones who had to manage the changes in clinic operations and health services, and most leaders were cognizant of the overwhelming stress these changes created for their workers. These stressors included having to keep up with the rapidly changing medical and public health information at the start of the pandemic, assuming the new role of providing technological support to patients for televisits and dealing with patients who did not disclose previous COVID-19 exposure until they were face-to-face with the clinician or staff member, thereby increasing the risk of exposure for frontline workers. One leader reflected that many healthcare workers had to contend with multiple exposures, which were also stressors:

….*our staff get[s] a double whammy. You know, they have to deal with everything that the general population is dealing with as far as concerns and anxieties over COVID plus they have to come to work in an environment where they’re at additional risk for exposure*.…[FQHC 2, Chief Medical Officer (CMO)]

Another major concern for healthcare workers was the fear of contracting COVID at work and transmitting it to their family members, especially family members who were elderly or had conditions that increased their risk for fatal infection.

In addition to work-related stressors, leaders discussed the personal stressors that their employees were experiencing. They consistently commented on the stress associated with having school-aged children at home, the lack of childcare, and, for some healthcare workers, having to choose between coming to work (and leaving children unattended) or staying home. For those who were able to work remotely, the challenge was managing their full-time work alongside supporting school-age children’s remote education. Leaders also acknowledged that school closures had differential impacts on healthcare workers depending on their socioeconomic status:

*I have a [young child] whose school shut down for two months, but, as I mentioned, I was fortunate enough to be able to continue to pay tuition to the school to keep her spot and also pay for someone to come to my home three days out of the week to continue instruction with her. But I realize that that’s a privilege that many of my staff don’t have, and definitely empathize with the fact that people were making hard decisions around what they had to do with their families*.[FQHC 5, CMO]

At the other end of the spectrum were the healthcare workers who lived alone, and for whom working remotely was socially isolating.

### 3.3. Organizational Support for Frontline Workers

When asked about how the leadership was supporting frontline workers, some interviewees described formal support services; others were only able to inform their workers of the resources available in the community. Two leaders representing different organizations discussed the use of the Families First Coronavirus Response Act (FFCRA), which provided funds for the organization to give their employees additional paid time off to support their families. One interviewee also shared that workers were able to use the Family and Medical Leave Act (FMLA) options. Two organizations also implemented “mental health days” or “respite time” for their employees. The mental health day was for employees who worked onsite at least 80 percent of the time; these workers were provided a free day that they negotiated with their supervisor. The organization that used “respite time” allowed everyone four hours a week to do whatever they wanted to *“help relax and calm themselves during the pandemic”*. Another form of support included the provision of mental health resources for employees. Two leaders described onsite counselors who were available for meditation guidance and other stress management techniques. There was also a specific assistance program offered through one organization’s human resources department to help workers with home life stressors. Only two leaders discussed a sick leave policy for employees who tested positive for COVID. Aside from their general sick leave, those who tested positive received 14 days of paid leave. One leader discussed a pending childcare grant that would be available to all qualifying employees.

Besides the various forms of formal support, most healthcare leaders felt unsure of how to best support their employees and were reconciled to be responsive and of help in any way possible. This informal support included being more available to listen to employee concerns. In addition, many leaders talked about their intentional efforts to increase communication with staff as a group through daily check-ins, meetings two to three times a week, and weekly rounds. These meetings were not only to discuss work-related issues but also to make sure that “*everyone is okay just dealing with the stress, anxiety of it all, childcare issues, family issues along with work and trying to help staff manage [work] expectations*.” [Healthcare organization 2, Director].

Increased operational flexibility was another support strategy. Leaders described being flexible with meeting times to accommodate their employees’ needs. There was also some flexibility with work hours, allowing people to conduct their work in the evenings so that they could focus on children or other family needs during the day. Others noted their increased flexibility with paid time off—allowing for longer time periods and expediting the approval process. However, being able to get more leave did not always include being paid for the time off.

Healthcare leaders also referred employees to external community resources to use as needed. Most of these resources were for psychosocial support such as free meditation classes and online resources for stress management. Another leader identified support from a local sports and recreation club that was offering free childcare for healthcare workers and communicated this resource to her staff.

Finally, a few leaders discussed their efforts to demonstrate employee appreciation. These efforts focused on activities to increase morale such as virtual happy hour and providing a “live” DJ virtually. Another organization modified their employee appreciation week such that each day they received a different reward, such as a free breakfast or lunch, lottery tickets, and an extra day off. When asked if there was anything else they wished they could do for their frontline workers, most leaders believed they had already maximized their options.

## 4. Discussion

The present study elucidates the impact of COVID-19 on clinic operations, organizational management, patient care, and frontline staff from the beginning of the pandemic. Interviews with healthcare leaders of primary care practices and community health centers revealed the uncertainty and fluidity of the situation in the early days of the pandemic—a time during which there was no federal response and the science about COVID-19 was still emerging [12]. This environment forced leaders to make rapid but sometimes temporary consequential decisions with limited information, as well as decisions that would impact the delivery of health services to high-need patients. Common decisions included a shutdown of services and transitioning to both telework and telehealth. Furthermore, organizations adjusted staff roles to allow them to provide services within the community to address social determinants of health potentially related to patient outcomes.

Support for frontline workers was also a significant challenge for the healthcare leadership. Although leaders had some understanding of the stressors frontline workers were experiencing or likely to experience, they were faced with having to assess and meet both the needs of the clientele in addition to the emerging needs of their employees for whom there were limited formal options for providing support. Ultimately, healthcare leaders reported stepping up to provide primarily psychosocial support to their frontline workers. The leadership’s behaviors of being more readily available to listen to workers’ concerns and flexible regarding accommodations indicated their empathy and willingness to be helpful. When considered alongside recommendations for creating resilient organizations in times of crisis [16], some of the healthcare leadership responses, however informal, aligned with the recommendations for providing psychosocial and mental health support. At the same time, their responses indicated unmet opportunities for more extensive support, even amid uncertainty. Although the leadership recognized work-related stressors, their support of frontline workers could not address the underlying stressors associated with their employment. Facilitative leadership in which multiple levels of workers are included in decision-making is associated with less work-related stress [17]. However, in this study, healthcare leaders did not mention including midlevel managers or frontline workers in the decision-making around changes in clinic operations.

As may be expected given the rarity of the pandemic, leaders were unclear about the elements of an appropriate systems-level response for supporting frontline workers. They were also not aware of other steps they could have taken to implement these elements to support frontline workers. When asked about other supports they wished were available to support their frontline workers most leaders believed that they were doing all that they could, and some believed it was sufficient. Indeed, the pandemic was unprecedented, financial resources were limited, and these healthcare leaders were managing multiple issues related to clinic operations. These findings highlight the utility of having a designated person to oversee employee well-being, brainstorming, and working with employees to identify helpful strategies and resources [16]—such efforts require advanced planning and therefore should be pursued prior to a crisis. Some practices to address work-related stressors that may not have created additional costs included informing workers on strategies to decrease their risk of “taking the virus home,” facilitating peer support groups, and suspending nonessential tasks. Partnering with employees to plan and execute such activities could also contribute a sense of psychological safety and decrease feelings of work-related stress.

Our findings align with other ongoing research during the period of COVID indicating that healthcare organization leaders remain challenged in supporting their workers. For example, respondents to the Larry Green survey indicated concerns about their organization’s capacity to meet the needs of their patients and a lack of policies or guidance to address their concerns [12]. The more recent introduction of long-haul COVID as a persistent health issue was acknowledged as an additional stressor, with over 60% of respondents expressing concern about the ability of the primary care workforce to address this new challenge. Although the respondents to the Larry Green survey may not represent clinicians and other healthcare workers in CHC settings, these findings underscore the need for support from healthcare leadership to improve conditions for its workforce and help primary care centers prepare for and address new developments in the ongoing pandemic.

Our findings also underscore an opportunity to address all components of the Quadruple Aim for improving quality care. In addition to the original aims of “enhancing patient experience, improving population health, and reducing costs,” improving the work life of healthcare professionals has been added as a fourth aim in recognition of its importance in achieving the quality targets [18]. However, there appear to be few incentives for improving the work-life balance of healthcare workers. Considering this, policy recommendations to support a healthy workforce should include publicly available staff satisfaction ratings for health organizations receiving federal dollars, development of workforce measures to assess staff well-being, and requiring organizations to describe their approach to reduce employee burnout on program evaluations for grant-funded programs, such as the Ryan White HIV/AIDS Program or the Federally Qualified Health Center Program. These actions would amplify the importance of a healthy workforce and enable organizational leaders to incorporate the impact on frontline workers in their decision-making, particularly in times of disasters. Furthermore, these are practical steps that can be taken to meet the systems-level recommendations from the National Academy Report on clinical burnout [19]. To achieve resiliency and well-being among frontline workers, targeted investments are needed. The funding opportunities provided with the American Rescue Plan through the Health Resource and Services Administration (HRSA) is one such example [20].

### Limitations

This study has several limitations. First, our sample included participants mainly from one state; as such, our findings may not capture the spectrum of organizational leadership experiences in response to the pandemic. Second, the leaders who participated in the interviews were those who were most involved in the implementation of the pragmatic trial in which this data collection was embedded. It may be that other leaders within the organization had greater levels of involvement in COVID responsiveness. Finally, the pandemic has remained a fluid situation; therefore, participant responses represent the time at which the interviews were conducted.

## 5. Conclusions

The respondents in this study reiterated the many challenges that CHCs and other primary care practices within larger health systems had experienced due to the pandemic. Amid the many operational changes, the healthcare workforce was challenged to adapt and deliver services with little organizational support to address the new pandemic-related work and personal stressors. Given the ongoing pandemic, continued work with the healthcare workforce will provide additional perspectives on support from their respective organizations. In the meantime, our findings underscore the need for more attention and resources to support healthcare workers in primary care settings especially during emergencies such as COVID-19.

## Figures and Tables

**Table 1 ijerph-19-03310-t001:** Themes and sample quotes from healthcare leaders.

**Theme: Restructuring Clinic Operations**
“The impact on our revenue has been really very difficult and we’ve come close a couple of times to having to make decisions about furloughs and layoffs but so far we haven’t had to do that. We are, you know, running a significant deficit at this point and we haven’t decided how we’re going to handle that before the fiscal year ends. But that is the biggest challenge.” [FQHC 2, Chief Medical Officer]
“We had to… because we implemented screening, where that doesn’t exist in our typical health center operations, and so just thinking about who was going to do that, who was available. So, we initially had our community health workers doing a lot of the temperature checks and asking the screening questions and would supplement that with the MAs [medical assistants].” [FQHC 5, Chief Medical Officer]
**Subtheme: Implementation of Telemedicine**
“We tried to make decisions that would not be a negative impact on those groups of individual patients, and so, of course, we went to telehealth and we didn’t know how that would work, but it’s working well. Of course, a few bumps in the beginning, but that turned out to work well, especially dealing with our behavioral health patients and protecting our newborns and our pregnant women. So, we just, you know, we didn’t allow sick patients to access the building. When I say sick, I mean someone who is complaining of earache, of sore throat, one who had a fever we did not permit to access the facility, so we did not schedule any of those sick patients at the same time with our well patients, and also we had to consider our newborn to two-year-olds to make sure they were receiving their vaccines on time. It was a lot to consider.” [FQHC 3, Director]
“…the transition to telemedicine actually helped us, because what we were able to do is limit the amount of staff that we had in the clinic at any given times, because there were folks who worked remotely, so that also helped us. It kind of worked together for the greater good, having to have people home and then work remotely. And then, we had less need for people to be on site, less PPE usage.” [FQHC 4, Director]
**Subtheme: Keeping Up with Patients’ Needs**
“I guess with the pandemic and so many places were closed maybe resources not being available to those patients. It was a trying time for those patients that needed medication assistance or food assistance or shelters. I think it’s challenging where we are anyways any given day, but when you throw the pandemic into it on top of it, I think it was more challenging for them.” [FQHC 5, Director]
“…we have worked hard with our counties on getting testing available. So we’ve run a couple of testing sites. We ran one at a school nearby, and we ran one at a church near one of the migrant camps that we have, and so availability of testing has been a challenge for folks with transportation to be able to get to places. We will pretty much test anybody if they come and want to have a test, but we’ve got to get that communication to them.” [FQHC 3, Chief Operations Officer]
**Theme: Stressed Frontline Workers**
“I think that they have had personally deaths in their families. They’ve had their own sickness. They’ve had children at home that have been sick and afraid. They’ve had violence in their families and around them. I think that everybody’s been impacted to a place of almost feeling numb.” [Healthcare organization 1, Chief Executive Officer]
“Them not having childcare, that’s a nightmare. And then they’re not even sure if they can keep their job then because they’re being told you need to come into work…. people who live with somebody, like anybody, like there was a girl I work with, wonderful CNA. And she has diabetes and some other conditions. She was like, “I know I’m at risk. I’m terrified.” And then another girl who talked to me who lives with her parents and her parents are elderly and immune compromised. And she was like, “If I bring this home to them I might kill them.” That’s terrifying.” [FQHC 5, Manager]
**Theme: Organizational Support for Frontline Workers**
“…everyone was given four hours a week of respite time, and that was four hours that they could do whatever they wanted to help relax and calm themselves during the pandemic… I think that was a great way that [the organization] showed its employees that we really care about you as a whole person. Yes, you’re doing a great job, but we also want you to take care of yourself physically and emotionally.” [FQHC 5, Director]
“We’re being much more flexible with work hours. And so if people need to work in the evening a little bit to do documentation while they’re working with their children during the day, we’re trying to be much more flexible with that. If someone does test positive they receive 14 days of paid leave. They don’t have to take sick time for that.” [Healthcare organization 2, Manager]

## Data Availability

Deidentified transcripts may be available upon request.

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
