# Peer review of "Healthcare Leadership Perspectives on Supporting Frontline Workers in Health Center Settings during the Pandemic"

_ijerph, 2022, doi:10.3390/ijerph19063310_

Round 1

Reviewer 1 Report

Dear Authors,

Thank you for taking up a very interesting topic of „Healthcare leadership perspectives on supporting frontline workers in health center settings during the pandemic”.  

The study was designed and carried out correctly. The results were presented clearly and properly discussed.  

In the introduction, the authors did not refer to the literature on leadership, but I do not see it as a weakness of the manuscript. As indicated by the authors, the purpose of this study was to understand and describe how organizational healthcare leaders have supported frontline workers in ambulatory settings during the pandemic. This goal has been achieved.

The only thing worth adding to the text is one or two sentences on the details of the analysis of the obtained research data, such as how many minutes was the average length of one interview (and total), how many pages was the transcription, etc. This type of information usually appears in papers that refer to research based on a qualitative perspective and interviews. However, I do not see this as a necessity. I leave the decision to the authors.

Best regards,
The reviewer.

Author Response

The study was designed and carried out correctly. The results were presented clearly and properly discussed.  

Response: Thank you so much.

In the introduction, the authors did not refer to the literature on leadership, but I do not see it as a weakness of the manuscript. As indicated by the authors, the purpose of this study was to understand and describe how organizational healthcare leaders have supported frontline workers in ambulatory settings during the pandemic. This goal has been achieved. Response: Thank you for this great point, in our review of the literature on CHC leadership responses we presented what we found (paragraph # 4 in introduction), leadership perspectives thus far reflect inpatient settings – often a very different context.

The only thing worth adding to the text is one or two sentences on the details of the analysis of the obtained research data, such as how many minutes was the average length of one interview (and total), how many pages was the transcription, etc. This type of information usually appears in papers that refer to research based on a qualitative perspective and interviews. However, I do not see this as a necessity. I leave the decision to the authors.

Response: Thank you so much, we have added these details (lines 104 – 106).

Reviewer 2 Report

This is a qualitative study on how leaders or supervisors tried to adjust to the crises caused by the pandemic and how they tried to support frontline workers caring for underprivileged patients.

  1. I am not sure the conclusion of the abstract is correct. You write. “While the leadership expressed recognition of the many personal and work-related stressors experienced by frontline workers, the options and opportunities for supporting these workers remained limited.” The second part of the conclusion is not prepared by the result section of the abstract. I guess in theory, there are many options but as resources are limited in practice this options are difficult to apply. The last sentence of the conclusion in the manuscript sound more convincing to me. Please consider.
  2. The study is part of a larger study. Are there other publications form the larger study describing the methods? If yes, please refer to these publications.
  3. Line 113: Sentence: All participants consented to participate in the extended interview that addressed 113 responses related to COVID workplace adaptation. I wonder if this is a result or an inclusion criterion, which should be moved to the method section?
  4. The participants were asked how the leadership was supporting frontline workers. A wide range of supportive measures is described, which is very interesting for a reader with another background as healthcare system. However, were participants also ask whether the support was helpful for the frontline workers? This did not become clear to me. Alternatively, is it that the participants perceived more need for support than they were able to provide? I think that could be better elaborated.

Thank you for the chance to read this interesting paper.

Author Response

  1. I am not sure the conclusion of the abstract is correct. You write. “While the leadership expressed recognition of the many personal and work-related stressors experienced by frontline workers, the options and opportunities for supporting these workers remained limited.” The second part of the conclusion is not prepared by the result section of the abstract. I guess in theory, there are many options but as resources are limited in practice this options are difficult to apply. The last sentence of the conclusion in the manuscript sound more convincing to me. Please consider. Response: Thank you so much for sharing this perspective. We have taken your suggestion and changed the last sentence of the Abstract.
  2. The study is part of a larger study. Are there other publications form the larger study describing the methods? If yes, please refer to these publications. Response: Thank you so much, we have referred to the main publication that we have – citation #15.
  3. Line 113: Sentence: All participants consented to participate in the extended interview that addressed 113 responses related to COVID workplace adaptation. I wonder if this is a result or an inclusion criterion, which should be moved to the method section? Response: Thank you, we have followed your suggestion and moved this to line #97 in the Procedures section.

The participants were asked how the leadership was supporting frontline workers. A wide range of supportive measures is described, which is very interesting for a reader with another background as healthcare system. However, were participants also ask whether the support was helpful for the frontline workers? This did not become clear to me. Alternatively, is it that the participants perceived more need for support than they were able to provide? I think that could be better elaborated. Response: Thank you for this point – we asked participants (line 280) whether or not there was anything else they could do or wished they could do for their workers and the majority believed that they had maximized their options – we have added this point to the discussion (line 317). Only one participant believed that the could and should be doing more to support staff.